# 4-Hydroxybenzoic Acid as an Antiviral Product from Alkaline Autoxidation of Catechinic Acid: A Fact to Be Reviewed

**DOI:** 10.3390/plants11141822

**Published:** 2022-07-11

**Authors:** Silvana Alfei, Debora Caviglia, Susanna Penco, Guendalina Zuccari, Fabio Gosetti

**Affiliations:** 1Department of Pharmacy (DIFAR), University of Genoa, Viale Cembrano, 4-16148 Genoa, Italy; zuccari@difar.unige.it; 2Department of Surgical Sciences and Integrated Diagnostics (DISC), University of Genoa, Viale Benedetto XV, 6-16132 Genoa, Italy; 3Department of Experimental Medicine, University of Genoa, Via Leon Battista Alberti, 2-16132 Genoa, Italy; susanna.penco@unige.it; 4Department of Earth and Environmental Sciences (DISAT), University of Milano-Bicocca, Piazza della Scienza, 1-20126 Milano, Italy; fabio.gosetti@unimib.it

**Keywords:** (+)-catechin, catechinic acid (CA), alkaline autooxidation of CA (AOCA), 4-hydroxybenzoic acid (4-HBA), antiviral activity, ultra-high-performance liquid chromatography (UHPLC), thin-layer chromatography (TLC), ultraviolet–visible spectroscopy (UV–Vis), attenuated total reflection Fourier transform infrared (ATR–FTIR), principal components analyses (PCA)

## Abstract

The dark brown mixture resulting from the autooxidation of catechinic acid (CA) (AOCA) has been reported to possess antiviral activity against Herpes Simplex Virus 1 and 2 (HSV-1 and HSV-2). Unfortunately, the constituents of AOCA were not separated or identified and the compound(s) responsible for AOCA’s antiviral activity remained unknown until recently. Colorless 4-hydroxy benzoic acid (4-HBA) has been reported as the main constituent (75%) of AOCA, and as being responsible for its antiviral activity. The findings seemed not to be reliable because of the existence in the literature of very different findings, because of the high concentration that was attributed to the supposed 4-HBA in the dark mixture, and because of the absence of essential analytical experiments to confirm 4-HBA in AOCA. Particularly, the AOCA chromatograms highlighting a peak attributable to 4-HBA, using commercial 4-HBA as a standard, is missing, as well as investigations concerning the antiviral activity of marketed 4-HBA. Therefore, in this study, to verify the exactness of the recent reports, we prepared CA from catechin and AOCA from CA, and the absence of 4-HBA in the mixture was first established by thin-layer chromatography (TLC), and then was confirmed by UHPLC–MS/MS, UV–Vis, and ATR–FTIR analyses. For further confirmation, the ATR–FTIR spectral data were processed by principal components analysis (PCA), which unequivocally established strong structural differences between 4-HBA and AOCA. Finally, while the antiviral effects of AOCA against HSV-2 were confirmed, a commercial sample of 4-HBA was completely inactive.

## 1. Introduction

Infusions obtained with the leaves of *Combretum micranthum* G. Don (Combretaceae) are commonly used in the folk medicine of West Africa, for the treatment of various diseases [1,2]. As reported, the main constituents in this species are polyphenolic compounds, including *O*-glycosylflavones, *C*-glycosylflavones, flavans [1,3,4], and flavan-piperidine alkaloids (kinkéloids) [2].

In 1993, and later in 2000, it was reported that while a freshly prepared solution of the methanolic extract of *C. micranthum* leaves was inactive against the HSV-1 and HSV-2 viruses, the one-week-aged extract or the extract subjected to alkaline autoxidation was, instead, particularly active against the same viruses [5,6]. In this regard, it has been hypothesized that aging, catalyzed in an alkaline environment, combined with heating at temperatures above 90 °C, would have been responsible for the formation of active compounds from the inactive precursors present in the extract [5]. Considering that catechins are the main constituents of the methanolic extract and that, in a hot alkaline solution, (+)-catechin undergoes rearrangement to catechinic acid (CA) (Figure 1) [7], Ferrea and coworkers prepared CA from pure (+)-catechin [5].

Subsequently, by the same group, the synthetized CA was oxidized in an alkaline oxygen-saturated solution at 100 °C, and the dark brown complex reaction mixture (AOCA) was chemically and biologically investigated [5]. Almost comparable antiherpetic activity and very similar IR and UV–Vis spectra of the alkaline oxidized methanolic extract and AOCA were observed, thus establishing that the oxidation products of CA were the main ones responsible for the antiviral activity in the oxidized extract [5]. Wishing to develop from AOCA a new antiviral agent, active against the viruses of the Herpes family, Ferrea and colleagues subsequently tried to isolate and identify the main component of AOCA, but unfortunately, they only succeeded in dividing AOCA into several fractions, all endowed with similar antiherpetic activity.

In a recent study, the main constituent (75%) of AOCA was isolated and named compound **2** [8]. It was identified as the colorless and well-known 4-hydroxy benzoic acid (4-HBA), and it was reported to be responsible for the antiviral activity of AOCA, as **2** had been found to be active on tomato brown rugose fruit virus (ToBRFV). This finding turned out to be very different from what was already widely reported in several specific articles [9,10,11,12,13]. In this regard, the structural identity recently attributed to **2** was validated, since 4-HBA is commonly produced by the thermolytic degradation of some flavonoids [14,15]. However, the thermolytic conditions under which this occurs are considerably more drastic than those used to produce AOCA. Moreover, it is known in the literature that 4-HBA derives only from flavonoids structurally different from catechin [14,15]. In particular, catechin, encompassing two catechol phenyl rings both having two hydroxyl groups, it is unlikely to thermolytically provide 4-HBA, which possesses only one hydroxyl group. In fact, it is reported that 4-HBA derives from flavonoids with at least one phenyl ring having only one hydroxyl function, and whose structure can precisely degrade to 4-HBA [14,15]. In addition, while many flavonoids are known to degrade into simpler phenolic molecules, such as protocatechuic acid, phloroglucinol, 4-HBA, and caffeic acid, none of these compounds have been observed in the alkaline degradation of flavonoids that initially provide CA, such as catechin [16]. In practice, the CA resulting from the alkaline treatment of (+)-catechin has been shown to be sufficiently stable against further alkaline transformation, to allow only the combination of two or more CA molecules and, therefore, permitting mainly the formation of dimers, in which the nucleus of CA is maintained [9,16]. In fact, while, through the initial formation of a quinone derivative, hydroxycatechinic and dehydrocatechinic acids and their dimers have been described as the main constituents of the product of the alkaline oxidation of CA obtainable under conditions such as those used to obtain AOCA, 4-HBA has not been isolated so far [9]. Additionally, the other compounds described by Laks et al. [10], Ohara et al. [11], and Hashida et al. [12,13], as originating from the alkaline degradation of CA, recently were not detected together with compound **2** [8]. Anyway, we think that a simple experiment to confirm the structural identity (4-HBA) attributed to **2** could have been performed, in the form of testing a commercial sample of 4-HBA on ToBRFV, confirming or refuting its identical antiviral activity. Precisely for this purpose, we tested, as reported by Ferrea et al. in 1993, a commercial sample of 4-HBA on VERO cells infected with HSV-2 viruses and discovered that 4HBA was completely inactive. Therefore, also for this reason, the identity attributed to compound **2**, found active against ToBRFV, was unlikely, especially at the high concentration reported (75%). In addition, a chromatogram of AOCA that highlighted the presence of a peak attributable to 4-HBA, using a commercial sample of 4-HBA as a reference compound, has not been provided [8]. Therefore, we undertook this study to verify the accuracy of the identity attributed to the molecule (**2**) that the authors were able to separate [8]. To this end, from (+)-catechin, CA was prepared, completely characterized, and oxidized to AOCA, according to procedures reported years ago [5,7] and re-performed in this recent study [8], and the possible presence of 4-HBA in the mixture was investigated by different analytical techniques. In particular, the absence of 4-HBA in the mixture was first established by thin-layer chromatography (TLC), and then was confirmed by UHPLC–MS/MS, UV–Vis, and ATR–FTIR analyses. Additionally, the ATR–FTIR spectral data were processed by principal components analysis (PCA), which unequivocally established strong structural differences between 4-HBA and AOCA, thus further confirming that AOCA does not contain 4-HBA. Finally, the antiviral effects of AOCA against HSV-2 were investigated and confirmed, thus establishing that the AOCA prepared by us, and not containing 4-HBA, is identical to the one recently reported and testified to contain 4-HBA, but whose activity against HSV-2 had not been verified.

## 2. Results and Discussion

### 2.1. Preparation of Catechinic Acid (CA)

CA was prepared as previously described [5,7,8,13], starting from commercial (+)-catechin according to Figure 1, in which the numbering of atoms of catechin and CA has been reported, to facilitate an understanding of the mechanism of the reaction and to clarify the assignments reported in the peak lists of NMR analyses (Materials and Methods section). We make note that the correct structure of CA presents the catechol ring linked to C-6 and not to C-8 as recently reported [8]. In fact, C8 should have two diastereotopic proton atoms, i.e., Ha and Hb and not only one, proton atom, just as herein observable in Figure 1 and Figure 1.

The reaction was followed by TLC, which allowed us to highlight the disappearance of the spot of (+)-catechin at Rf. 0.82 and the appearance of that of CA at Rf. 0.26. An image reporting the TLC of (+)-catechin and of the crude CA, of the isolated CA, and of three samples of CA recrystallized by acetone/Et_2_O is available in Appendix A. Specifically, it has been reported [7,13] that the basic treatment of (+)-catechin at 25 °C provides enantiomeric epicatechin by epimerization at the C-2 position of the pyran ring, whereas CA is obtained when the same treatment is conducted at 100 °C [7,13,17]. These reactions are supposed to proceed both through the basic catalyzed cleavage of the pyran ring to give the quinone-methide intermediate, followed by the reclosure of the pyran ring affording epicatechin, or by the nucleophilic attack of the π doublet between C-7 and C-8 of the phenyl ring at the C-2 position of the quinone-methide intermediate to afford CA (Figure 1). A radical reaction mechanism was proposed because these reactions are inhibited by the exclusion of oxygen [13,18]. Differently, Ohara and Hemingway showed that at an intermediate temperature (40 °C), the di-aryl-propanol-catechin acid dimer is produced together with catechin acid [11], and other unidentified compounds are produced in low yields [13]. Chemically, CA is an enolic form of 6-(3,4-dihydroxyphenyl)-7-hydroxybicyclo [3.3.1]-nonane-2,4,9-trione [13]. We obtained crude CA as a dark brown oil, and upon purification by fast filtration on a silica gel plug, followed by crystallization, we achieved highly pure CA as pale orange crystals (70% isolated yield) (Appendix A). Notably, extremely neat CA was achieved, without recovery via high-cost and long-term semi-preparative HPLC, which uses large amounts of solvents and costly apparatus, as recently reported [8]. In our opinion, special purification techniques should be used when strictly necessary to lower costs and respect the environment. The structure of the obtained CA, which we isolated in a simple, cheap, and fast manner, was confirmed by ATR–FTIR (Figure 2) and ^1^H and ^13^C NMR spectroscopy (Appendix A), while its purity was assessed first by TLC (Appendix A), and then by UV–Vis and UHPLC–MS/MS analyses (Figure 3 and Figure 4), and finally further confirmed by elemental analyses.

The results obtained by us from UV–Vis and ATR–FTIR analyses conformed to those reported by Sears et al. (UV–Vis and FTIR) [5] and Kennedy et al. (UV–Vis) [18]. The slight shift in the λ_max_ of the UV–Vis analysis of CA (267 nm vs. 285 nm) is probably due to the different solvents used in the acquisition (MeOH vs. H_2_O). In particular, in the UV–Vis spectra of CA and (+)-catechin (Figure 3), peaks with very similar shapes were observed, due to the strong structural correlation existing between the two molecules. On the contrary, different values of λ_max_, indicating the presence of different chromophores and confirming the pureness of CA, were detected. Similar to Sears, we observed decomposition for temperatures over 190 °C, and elemental analysis provided results in the range of acceptability, further confirming the pureness of CA. Concerning the NMR spectra of CA, the ^1^H NMR signals observed by us in the spectrum acquired in CD_3_OD were in accordance with those recently reported [8], which are in turn in compliance with those reported previously [13], with minor differences probably due to the different solvents [13] and the instruments’ potency [8,13]. On the contrary, the signal of C-4 in the ^13^C NMR observed by us was at a chemical shift considerably lower (higher fields) than that reported by Hashida [13] and recently provided [8] (152 ppm vs. 182.08 [13] or 189.1 [8]), but in accordance with the value reported by Navarrate et al. and with the keto–enol structure of CA [17]. Nonetheless, the higher values observed by the other authors are also acceptable, since they can be justified by the existence of an enol–enol tautomerism in the crystalline keto–enol structure of CA [19]. Figure 4 shows both the UHPLC–MS/MS analysis of CA and of (+)-catechin, having Rt = 0.87 min and 1.98, respectively, as well as the level of pureness of CA (only 4% of residual (+)-catechin is present in the CA sample), thus further confirming the quality of the CA prepared by us.

### 2.2. Preparation of AOCA and Search for the Presence of 4-HBA

Starting from CA, we prepared AOCA by reproducing exactly the protocol developed in the past by Ferrea et al. [5] and recently reproduced [8]. AOCA was obtained as a dark brown solid, as observable in Appendix A. Additionally, we completely characterized AOCA in terms of physicochemical analyses, by TLC, ATR–FTIR, UV–Vis and UHPLC–MS/MS, which were not carried out so far.

In particular, we prepared AOCA and carefully analyzed it by several analytical techniques to unequivocally verify the accuracy of the structural identity attributed to compound **2** (identified as 4-HBA) recently isolated as the main constituent of AOCA and reported as responsible for its good antiviral activity [8]. However, compound **2** has been shown to be active on a completely different type of virus (i.e., on ToBRa, a naked RNA virus) compared to the one on which the AOCA was originally found to be active (i.e., HSV-1 and HSV-2, DNA viruses, coated with an envelope) [5]. This study was motivated by the lack of literature reporting 4-HBA as a constituent, at least in trace amounts, of the products of the alkaline oxidation of catechin or CA [9,10,11,12,13], and by the presence of several literature studies that describe the remarkable stability of the CA nucleus under the alkaline oxidative conditions [16]. Moreover, since we believed that analyses such as HPLC or at least TLC, highlighting the actual presence of 4-HBA in AOCA, were necessary, they were herein carried out by us. Furthermore, the results from NMR analysis carried out on compound **2** are not totally in accordance with those known for commercial samples of 4-HBA. In particular, even if, in the ^1^H NMR spectrum of **2**, signals compatible with the AA’XX’ system of the *p*-disubstituted phenyl ring of 4-HBA are observable at 7.87 ppm (d, 2H, J = 8.00 Hz, CH (2/6)) and 6.82 ppm (d, 2H, J = 8.01 Hz, CH (3/5)), other, not negligible signals are present at 0.5–1.5 ppm (particularly 0.8 and 1.5 ppm), which are missing in the spectrum of a commercial sample of 4-HBA (Figure 5).

For the precise assignment of the signals of the AA’XX’ system observable in the ^1^H NMR spectrum of **2**, HSQC and HMBC 2D experiments have been reported [8]. In an HSQC spectrum, a ^13^C spectrum is displayed on one axis and a ^1^H spectrum is displayed on the other axis, and cross-peaks show which proton is attached to which carbon considering only C-H (only one-bond coupling). Meanwhile, two-, three-, or sometimes even four-bond couplings are signaled in an HMBC spectrum. In particular, H-C-C or H-C-C-C, or even H-C-C-C-C, and not H-C, are visible. Regarding this, in the HSQC spectrum of **2**, one-bond couplings between the proton atom(s) at 1.5 ppm with carbon(s) having signals in the range 20–40 ppm are visible, which are not compatible with the structure of 4-HBA [8]. Furthermore, since it was assumed that **2** could be 4-HBA, the ^1^H NMR spectrum was acquired in deutero methanol (which does not allow one to see the signals of exchangeable protons, thus losing valuable information) rather than in deutero dimethyl sulfoxide. The dimethyl sulfoxide spectrum would have instead allowed to confirm or refute the presence of the COOH and OH groups of 4-HBA.

Finally, differently from 2 (identified as 4-HBA) [8], a sample of commercial 4-HBA tested by us on VERO cells infected by HSV-2 was not active.

Based on these considerations, to verify the actual presence of 4-HBA among AOCA constituents, we examined the AOCA prepared by us, performing different analytical techniques essential to confirm the identity assigned to **2**. In the following section of this article, the performed analyses, the results, and a brief discussion are given.

#### 2.2.1. TLC Analyses of AOCA and 4-HBA

Employing aluminum-backed silica gel plates and detecting the spots at 254 nm and 365 nm, TLC analyses were carried out on AOCA and commercial 4-HBA, taken as a reference compound, using a mixture EtOAc/MeOH/AcOH 9/1/0.2 *v/v/v* (mixture A) as an eluent. Images of TLC are provided both in Figure 6a (AOCA on the left side of the TLC, 4-HBA on the right side) and in Section S2 of SM (Appendix A). Additionally, in Figure 6b, the TLC profile of AOCA obtained using MeOH 100% (mixture B) is included.

As observable in both Appendix A (254 nm) and Figure 6a (365 nm), there is no trace in AOCA of a spot corresponding to the Rf of 4-HBA, thus indicating the absence of 4-HBA in AOCA.

#### 2.2.2. UV–Vis Analyses of AOCA and 4-HBA

The UV–Vis spectra of AOCA and of a sample of commercial 4-HBA, taken as a reference compound, were acquired in MeOH, as described in the Materials and Methods section. The obtained spectra are reported in Figure 7. Additionally, in SM, an image showing the overlapped spectra of (+)-catechin, CA, AOCA, and 4-HBA is provided for the comparison of the UV–Vis profiles of all compounds considered in this study (Appendix A).

As Figure 7 and Appendix A show, in the AOCA spectrum (λ_max_ = 268 nm), no peak is detectable at λ_max_ = 257 nm, corresponding to 4-HBA absorbance, while, considering the recently reported high concentration of **2** in AOCA (75% *wt/wt*), if this had been the case, a significant absorption at λ_max_ = 257 nm should have been observed in AOCA. Additionally, both concerning the shape and the λ_max_, the UV–Vis spectra of CA and AOCA (Figure 3, Figure 7, and Appendix A), are very similar to each other, evidencing a strong structural correlation existing between the two molecules, and the presence of identical chromophores, suggesting that the nucleus of CA was maintained in AOCA. This agrees with what is widely reported, which asserts the presence of dimers of CA derivatives in the product of the oxidation of CA, rather than small phenolic molecules, such as 4-HBA. Furthermore, the very broad peak of AOCA suggests the presence either of several substances, as is typical of plant extracts [20], or of macromolecules, such as dimers and oligomers of CA [21], in different concentrations, rather than the presence of a small molecule such as 4-HBA at a concentration of 75%, whose UV peak appears much narrower.

#### 2.2.3. UHPLC–MS/MS Analyses of (+)-Catechin, CA, AOCA, and 4-HBA

To obtain the best selectivity and specificity, a mass spectrometer equipped with a triple quadrupole analyzer as a detector was employed. An SRM method was built, monitoring three transitions for each analyte in ESI negative ion mode.

UHPLC–MS/MS analysis of standard solutions of CA, 4-HBA, and (+)-catechin were carried out as described in the Materials and Methods section, and the related chromatographic peaks eluted at Rt of 0.87, 1.86, and 1.98, respectively (Figure 8a). Then, a UHPLC–MS/MS analysis of the AOCA sample was performed (Figure 8b): there were several peaks, and their identification was not within our scope, but, unequivocally, no peak, even in trace, related to 4-HBA (magenta trace) was detectable (LOD = 0.3 ng/mL), highlighting the incorrect attribution suggested for the isolated material, whereas CA (at Rt = 0.87) was determined at a concentration of 15.8 ± 1.1 mg/g of AOCA.

#### 2.2.4. Chemometric-Assisted ATR–FTIR Analyses of (+)-Catechin, CA, AOCA, and 4-HBA

ATR–FTIR spectra of commercial samples of (+)-catechin and 4-HBA, as well as of CA and AOCA prepared by us, were recorded, as described in the Materials and Methods section. The obtained spectra are available in Appendix A ((+)-catechin), Appendix A (CA), Appendix A (AOCA), and Appendix A (4-HBA) in Section S2 of SM. Moreover, in the subsequent Figure 9 are shown all the overlapped spectra (Figure 9a) and a magnification of a significant region of all the spectra (Figure 9b), to highlight similarities and differences among all the substances.

From a simple observation of the spectra of AOCA and 4-HBA, none of the intense bands specific to 4-HBA (1669, 1594, 1510, 1418, 1316, 1288, and 1241 cm^−1^), are detectable in the AOCA spectrum, while, if 4-HBA was **2**, isolated at 75% in AOCA [8], at least traces of such bands should have been detectable. Moreover, although merely by observing the ATR–FTIR spectra in Figure 9, the absence of 4-HBA in AOCA is unequivocal, we further confirmed this fact by processing the matrix of the ATR–FTIR spectral data of all samples using principal components analysis (PCA) [22,23,24,25,26]. PCA is a technique used in multivariate analysis (MVA), to process spectral data consisting of thousands of variables that need reduction to a smaller number of new variables, called principal components (PCs), which efficiently represent data variability in low dimensions. As a result, PCA provides score plots, where one component (e.g., PC1) is displayed vs. another (e.g., PC2), and where the samples under study assume specific positions (scores), forming groups of similar compounds. The position taken by each sample on the selected component can give predictive information on its chemical composition. Practically, compounds that are structurally similar will have similar scores and will be located close together, while compounds that are structurally different will have different scores and will be located distant from one another in the score plot. The PCA results allowed us to obtain the reciprocal positions of (+)-catechin (CAT), CA, AOCA, and 4-HBA occupied in the score plot of PC1 vs. PC2 (Figure 10).

All samples were well separated mainly on PC2 and reciprocally located at specific score values depending on their structural features. It was thus clearly evidenced that (+)-catechin, CA, and AOCA were positioned in the same square and at positive similar scores, thus testifying to their belonging to the same structural family, especially in the case of CA and AOCA. On the contrary, 4-HBA was positioned very distant from these compounds, at highly negative score values, thus confirming the strong structural differences between 4-HBA and AOCA, and the total absence of 4-HBA in AOCA.

### 2.3. Evaluation of the Antiherpetic Effects of AOCA on HSV-2 In Vitro

To verify that the AOCA prepared by us, and that of Ferrea et al. [5], who found AOCA active against HSV-1 and HSV-2, were comparable, thus validating our findings, we repeated, with AOCA on VERO cells the same tests carried out by Ferrea. In addition, hoping to highlight an antiviral activity of 4-HBA right on HSV-2, the same experiments were conducted with a commercial sample of 4-HBA. To this end, VERO cells were cultured as previously reported [5], while HSV-2 G strain was used to create viral infections in cells, as described by Ferrea and collaborators [5]. The results of these experiments reproduced those obtained by Ferrea et al., thus establishing that even the AOCA prepared by us possessed dose-dependent activity against HSV-2. On the contrary, the commercial sample of 4-HBA, tested in the same conditions, was completely inactive also at very high concentrations against HSV-2. In particular, in standard plaque forming assay experiments, AOCA showed significant activity, and approximately 100% reduction in plaque formation in comparison with un-treated cultures was obtained at 8.0 µg/mL.

## 3. Materials and Methods

### 3.1. Chemical Compounds and Instruments

(+)-Catechin analytical standard from Sigma-Aldrich^®^, of plant origin, was purchased from Merck (Darmstadt, Germany). All reagents were of analytical grade and purchased from Merck (Darmstadt, Germany). Solvents were purified by standard procedures, whereas reagents were employed as such, without further purification. The melting range of CA was determined on a 360 D melting point device, resolution 0.1 °C (MICROTECH S.R.L., Pozzuoli, Naples, Italy), while ^1^H and ^13^C NMR spectra were acquired on a Jeol 400 MHz spectrometer (JEOL USA, Inc., Peabody, MA, USA) at 400 and 100 MHz, respectively. Fully decoupled ^13^C NMR spectra were acquired. Chemical shifts were reported in ppm (parts per million) units relative to the internal standard tetramethylsilane (TMS = 0.00 ppm), and the splitting patterns were described as follows: s (singlet), d (doublet), t (triplet), q (quartet), and m (multiplet), and br was added for broad signals. Elemental analyses were performed on an EA1110 Elemental Analyser (Fison Instruments Ltd., Farnborough, Hampshire, England). Organic solutions were evaporated using a rotatory evaporator, Rotavapor^®^ R-3000 (Buchi, Cornaredo, Milan, Italy), operating at a reduced pressure of approximately 10–20 mmHg. Silicagel LC60A, 40–63 micron, purchased from Fluorochem Ltd. (Hadfield, Derbyshire, UK), was used for silica gel plugs.

Ultra-performance liquid chromatography coupled with tandem mass spectrometry (UHPLC–MS/MS) was employed in selected reaction monitoring (SRM) mode to identify the possible presence of 4-HBA in the AOCA sample. The LC/MS analyses were performed using the Nexera Liquid Chromatography Shimadzu (Kyoto, Japan) system, equipped with a DGU-20A3R degasser, two LC-30AD pumps, a SIL-30AC autosampler, a CTO-20AC column compartment, and a CMB-20 system. The system was interfaced with an LC-MS 8060 triple quadrupole mass spectrometer (Shimadzu, Kyoto, Japan) by an ESI ion source. The LC/MS data were acquired and processed with LabSolutions LCMS ver. 5.99 (Shimadzu Corporation, Kyoto, Japan) and LabSolutions Insight ver. 3.7 (Shimadzu Corporation, Kyoto, Japan) software.

### 3.2. Preparation of Catechinic Acid (CA)

Catechinic acid (CA) was prepared following the method described by Sears et al. [7]. In particular, (+)-catechin (Rf. = 0.82, ethyl acetate (EtOAc)/methanol (MeOH)/acetic acid (AcOH) 9/1/0.2 *v/v/v*) (1.0 g, 3.45 mmol) was added to a refluxing solution of 0.5% NaOH and stirred for 45 min. The dark solution was then cooled in ice and treated with Amberlite IR-120 resin (H+ form) (Merck, Darmstadt, Germany), and stirred for 1 h. The resin was then filtered off, and the solution was evaporated to dryness to give a dark brown oil, which was purified first by filtration through a silica gel plug (h = 20 cm, Ø = 2 cm) using a mixture of EtOAc/MeOH/AcOH (54/6/1.2 *v/v/v*) as an eluent. The solvent was then removed at reduced pressure (t = 60 °C) to obtain an orange solid, which was treated with ethyl ether (Et_2_O) and filtered. The obtained solid was further purified by crystallization (acetone/Et_2_O), washed with Et_2_O three times, and brought to a constant weight at vacuum (0.6927 g, 2.02 mmol).

*Catechinic acid (CA).* Isolated yield 69.2%. Rf. = 0.26 (EtOAc/MeOH/AcOH 9/1/0.2 *v/v/v*). Mp.: >190 °C (charring.). ATR–FTIR (ν, cm^−1^) 1745 (C = O); 1695, 1610, 1535. ^1^H NMR (CD_3_OD, 400 MHz, δ ppm) 2.06 (1H, ddd, H-8a, *J* = 14.23, 10.25, 2.73 Hz), 2.13 (1H, ddd, H-8b, *J* = 14.23, 2.84, 2.63 Hz), 3.39 (1H, dd, H-6, *J* = 10.25, 2.73 Hz), 3.76 (1H, dd, H-1, *J* = 2.84, 2.73 Hz), 3.90 (1H, d, H-5, *J* = 2.73 Hz), 5.12 (1H, td, H-7, *J* = 10.25, 2.84 Hz), 5.95 (1H, s, H-3), 6.69 (1H, d, H-6′, *J* = 8.52 Hz), 6.74 (1H, d, H-5′, *J* = 8.52 Hz), 6.80 (1H, s, H-2′).

^13^C NMR (100 MHz, δ ppm) 212.7 (C = O, C9), 199.5 (C = O, C2), 152.5 (COH, C4), 146.3 (COH, C4′), 145.8 (COH, C3′), 139.2 (C, C1′), 128.7 (CH, C5′), 115.8 (CH, C4′), 114.8 (CH, C2′), 104.5 (CH, C3), 69.3 (CHOH, C7), 73.8 (CH, C1), 58.8 (CH, C1), 40.7 (CH C5,6), 32.5 (CH2, C8). Anal. Calcd. for C_15_H_14_O_6_: C, 62.07%; H, 4.86%; O, 33.07%. Found: C, 62.37%; H, 4.90%; O, 33.00%.

### 3.3. Alkaline Autoxidation of CA

CA was then subjected to alkaline autoxidation, as previously described [5]. Briefly, in a three-necked flask equipped with a condenser, a thermometer, and a magnetic stirrer, an alkaline solution was prepared by dissolving 0.5 g NaOH in 7.5 mL of water. CA (0.5342 g, 1.84 mmoL) was added, and the dark brown solution was stirred for 30 min at room temperature, and then for another 30 min at 55 °C, and, finally, for 1.5 h at 100 °C. After cooling, the black solution was passed through Amberlite IR-120 (H+ form) (Merck, Darmstadt, Germany), an ion exchange resin column (h = 5 cm, Ø = 1 cm), and evaporated at reduced pressure to dryness, obtaining AOCA (0.5753 g) as a deep dark brown solid (AOCA).

### 3.4. TLC Analyses of (+)-Catechin, CA, AOCA, and 4-HBA

Thin-layer chromatography (TLC) was carried out employing aluminum-backed silica gel plates (Merck DC-Alufolien Kieselgel 60 F254, Merck, Washington, DC, USA), and detecting spots by UV light (254 nm or 365 nm), using a handheld UV lamp, LW/SW, 6W, UVGL-58 (Science Company^®^, Lakewood, CO, USA). The chromatographs were eluted in a closed glass developing chamber to keep the environment saturated with solvent vapors, using a mixture of EtOAc/MeOH/AcOH 9/1/0.2 *v/v/v* (mixture A) or 100% MeOH (mixture B).

### 3.5. UV–Vis Analyses of (+)-Catechin, CA, AOCA, and 4-HBA

The UV–Vis spectra of (+)-catechin, CA, AOCA, and 4-HBA were acquired using a UV–Vis spectrophotometer (HP 8453, Hewlett Packard, Palo Alto, CA, USA) equipped with a 3 mL cuvette. With the use of a microbalance, amounts of (+)-catechin, CA, AOCA, and 4-HBA were exactly weighted, solubilized in MeOH (50 mL), and appropriately diluted with the same solvent, obtaining solutions at the concentrations reported in Table 1. The obtained solutions were used for the analyses, recording the ultraviolet absorption spectra of all samples in the range 230–650 nm. Data of the analyses are included in Table 1.

### 3.6. UHPLC–MS/MS Analyses of (+)-Catechin, CA, AOCA, and 4-HBA Standard Solutions

The stationary phase involved a Raptor Biphenyl column (100 × 2.1 mm, 1.8 µm) acquired from Restek (Milan, Italy), and a Raptor Biphenyl UHPLC (5 × 2.1) was used as a guard column. The mobile phase was a mixture of (A) ultrapure water and (B) methanol, both with the addition of 0.02% of acetic acid, eluting under the following gradient conditions: 0.0–0.3 min 10% B, 0.3–5.0 min 100% B, 5.0–8.0 min 100% B, 8.1 min 10%. B. The final run time was 10.0 min. The flow rate was 0.45 mL min^−1^, the injection volume was 1.0 µL, and the oven temperature was set at 45 °C.

The ESI worked in negative polarity ion mode (NI) and the mass spectrometer was used in selected reaction monitoring (SRM) mode. The optimized instrumental parameters were set as follows: nebulizing gas flow 3.0 L/min; heating gas flow 10 L/min; drying gas flow 10 L/min; interface temperature 300 °C; DL temperature 250 °C; heat block temperature 400 °C. The unit mass resolution was established and maintained in each mass-resolving quadrupole by keeping a full width at half-maximum (FWHM) of approximately 0.7 u. The SRM transitions with the related dwell times and the optimized compound-dependent parameters, such as voltage potential Q1, Q3, and collision energy (CE), are reported in Appendix A.

### 3.7. Chemometric-Assisted ATR–FTIR Analyses of (+)-Catechin, CA, AOCA, and 4-HBA

FTIR spectra of commercial samples of (+)-catechin and 4-HBA, as well as of CA and AOCA prepared by us, were recorded in attenuated total reflection (ATR) mode directly on the sample on a Spectrum Two FTIR Spectrometer (PerkinElmer, Inc., Waltham, MA, USA). The spectra were acquired in triplicate for each compound in the transmittance scale. The acquisition was made from 4000 to 600 cm^−1^, with 1 cm^−1^ spectral resolution, coadding 32 interferograms, with a measurement accuracy in the frequency data at each measured point of 0.01 cm^−1^, due to the laser internal reference of the instrument. The frequency of each band was obtained automatically by using the “find peaks” command of the instrument’s software. The matrix created from the spectral data of all samples was subjected to principal component analysis (PCA) using PAST statistical software (Paleontological Statistics software package for education and data analysis, freely downloadable online, at https://past.en.lo4d.com/windows, accessed on 3 June 2022). In particular, the FTIR dataset of the spectra acquired for (+)-catechin, 4-HBA, CA, and AOCA was arranged in a matrix of n measurable variables. For each sample, the variables consisted of the values of absorbance (%) associated with the wavenumbers (3401) in the range 4000–600 cm^−1^. To simplify the system, we exploited PCA, which reduced the large number of variables to a small number of new variables, namely principal components (PCs). Chemometric analyses by PCA were performed on a matrix of data 4 × 3401, including a total of 13,604 original variables.

### 3.8. Antiherpetic Effects of 4-HBA and AOCA

African green monkey cells (VERO) (Flow Laboratories Ltd., Irvine, Scotland) were grown as previously reported [5], while HSV-2 strain G, kindly provided by Dr. R. Whitley (University of Alabama, Birmingham, AL), was stored at −70 °C until use. The antiviral activity of AOCA was determined following the procedure reported by Ferrea et al. [5]. In particular, monolayers of VERO cells in a 24-well Costar microplate (Bellco-Glass, Vineland, NJ, USA) were infected with 30 PFU of HSV-2 G strain with or without different concentrations of AOCA or 4-HBA. The plaque assay was performed as follows. Briefly, the virus was suspended either in RPM medium supplemented with 2% fetal calf serum or in six concentrations of AOCA or 4-HBA, and 1 mL aliquots of the suspensions were added to cell monolayers. After 1 h of absorption, the supernatant was aspirated and 1 mL of RPM medium containing 2% fetal calf serum and 0.5% human immune globulin and six different concentrations of AOCA or 4-HBA were added. Experiments were performed in quadruplicate. After 48 h of incubation at 37 °C in 5% CO_2_, the medium was removed, and the plaque formation was determined microscopically after methanol fixation and staining with GIEMSA solution. 

## 4. Conclusions

Compound **2** was recently isolated from the product of the oxidation of *Combretum* leaf extracts and catechinic acid (CA), namely AOCA. It was discovered to be active against ToBRFV and was identified as 4-HBA. Here, an unusual study has been reported, aimed at verifying the presence of 4-HBA among the constituents of AOCA (at a concentration above 50%) and its actual antiviral activity. To this end, we have prepared CA from (+)-catechin of plant origin and in turn we have prepared AOCA from CA. AOCA has been subsequently analyzed by performing TLC, UV–Vis, UHPLC–MS/MS, and chemometrically assisted ATR–FTIR experiments, unequivocally ascertaining that 4-HBA is not a constituent of AOCA. Differently from AOCA prepared by us and AOCA prepared by Ferrea et al., a commercial sample of 4-HBA was completely inactive when tested on HSV-2. From results of our study the identity of compound **2** remains unknown. With this work we have evidenced that the materials deriving from plants extraction are very complex mixtures of compounds, difficult to be separated and identified, so that detailed studies and verifications are necessary, as well as the development of new analytical methods.

## Data Availability

Data supporting the reported results are all presented in this manuscript or in the Appendix A included.

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
