# Peer review of "4-Hydroxybenzoic Acid as an Antiviral Product from Alkaline Autoxidation of Catechinic Acid: A Fact to Be Reviewed"

_plants, 2022, doi:10.3390/plants11141822_

Round 1

Reviewer 1 Report

The Authors improved mostly the manuscript content due to the Reviewers' comments.

I m recommending changing the title - please cross out "fake news" and try to reformulate it for more "scientific" than "media".

Please consider adding in conclusion, that for plant derivates detailed studies and verification are necessary i.e. due to diverse material as well as new analytical methods development. 

Author Response

The Authors improved mostly the manuscript content due to the Reviewers' comments.

We thank the Reviewer for appreciating the work made by us to improve our manuscript and for his help.

I m recommending changing the title - please cross out "fake news" and try to reformulate it for more "scientific" than "media".

“Fake news” has been crossed out and the title has been reformulated.

Please consider adding in conclusion, that for plant derivates detailed studies and verification are necessary i.e. due to diverse material as well as new analytical methods development.

The considerations proposed by the Reviewer have been added in the Conclusions (lines 539-541).

Reviewer 2 Report

Although the authors significantly improved their manuscript, the reviewer's main concerns are still remained: the manuscript does not contain enough novelty. Namely, this manuscript contains only two new results: i) the formation of 4-hydroxy benzoic acid during the autoxidation of catechinic acid, and ii) the antiviral activity of 4-hydroxy benzoic were disproved in this study.

In conclusion the manuscript, in its present form doesn’t meet the requirements of the journal Plants.

Author Response

Although the authors significantly improved their manuscript, the reviewer's main concerns are still remained: the manuscript does not contain enough novelty. Namely, this manuscript contains only two new results: i) the formation of 4-hydroxy benzoic acid during the autoxidation of catechinic acid, and ii) the antiviral activity of 4-hydroxy benzoic were disproved in this study.

In conclusion the manuscript, in its present form doesn’t meet the requirements of the journal Plants.

We thank the Reviewer for acknowledging the efforts made by us to improve our manuscript. However, it seems that our efforts have not been sufficient to make the Reviewer understand the purpose of our work and what the originality and relevance of our study consists of. The Reviewer himself defined our study “unusual”, and just since unusual, our work can be retained intrinsically innovative and original, thus opposing the Reviewer’s comment.

Additionally, the Reviewer is dramatically wrong when asserts that one of the new results contained in our work is “the formation of 4-hydroxy benzoic acid during the autoxidation of catechinic acid”, because our study demonstrated the exact contrary. Precisely, Iobbi et al have recently isolated an antiviral compound from the product of autooxidation of CA and have identified it as 4-hydroxibenzoic acid, thus asserting the formation of 4-hydroxy benzoic acid during the autoxidation of catechinic acid. Instead, we, finding this attribution unlikely and considering this news dangerous if not true, we conducted a systematic study to see if 4-hydroxybenzoic acid was actually present in AOCA and had antiviral properties, demonstrating the falsity of both revelations made by Iobbi et al.

Round 2

Reviewer 2 Report

Although this manuscript confirms only two new results (i.e., the formation of 4-hydroxy benzoic acid during the autoxidation of catechinic acid and the antiviral activity of 4-hydroxy benzoic were disproved in this study), it can be accepted for publication in the present form as these results may be of interest to readers of the journal Plants.

This manuscript is a resubmission of an earlier submission. The following is a list of the peer review reports and author responses from that submission.

Round 1

Reviewer 1 Report

This manuscript summarizes results of an unusual study aiming to revise some previous scientific results. Namely, i) the formation of 4-hydroxy benzoic acid during the autoxidation of catechinic acid, and ii) the antiviral activity of 4-hydroxy benzoic was disproved in this study. Evaluating the scientific significance of these two results it can be concluded that they are of little importance: the manuscript does not contain enough novelty. To strengthen the manuscript, the following should also be studied/confirmed:

1) Metabolites formed by the autoxidation of catechinic acid should be identified.

2) The metabolite responsible for the antiviral activity of the autoxidation products should be determined.

3) Instead of not purified catechinic acid (prepared as 70% isolated yield), a catechin acid with low impurity (i.e., purified by prep. HPLC or other method) should be used to perform autoxidation process: Impurities, which are present in not purified catechinic acid, can affect the formation of compounds during oxidation.  

In conclusion the manuscript, in its present form doesn’t meet the requirements of the journal Plants.

Reviewer 2 Report

The manuscript presented a study of CA and AOCA systems that can be delivered from plants. As I good understood, the experiments were performed based on the commercially available reagents not from plants.

For that reason, I m suggesting presenting the results in another kind of journal. If not, please clearly state it.

Nevertheless, the manuscript needs major revision. General remarks:

- for the first time I m reading the paper prepared in this way, where the Authors try to discuss unpublished results (see details below). Please have a look at your manuscript and change it

- the quality of the figures is poor (see details)

- the aim of the paper and the most important achievements should be shortly stated in the conclusions as well.

- the most important results about antiviral activity were not shown.

Detailed comments:

- Please change the title is not so informative (especially, the second part).

Abstract

- please rewrite the abstract in a way that presents more your studies. The storytelling is nice but more suitable for the introduction. Focus on your examinations, not on the others.

Introduction

- line 45 please check if the name in the bracket is correct)

- In line 69 you can cross out this subsection heading

- line 106 and next - these part is more suitable for discussion. Please rewrite it a little bit and try to avoid statements "it was very strange"... etc. in case of other previously published results. It is a little bit impolite, and not necessary (check lines 111 - 112 as well).

- some parts from the last section of the introduction should be repeated in the abstract (see the previous comment)

- I m suggesting extending the little bit the introduction by giving 4-5 current references (try to find some recent ones no older than 5-10years). My general suggestion is to extend a little bit the number of current references.

Results and discussion

-could you uniform the spectra i.e. use the same colors for each compound (it will be easier to recognize it in the manuscript). What is more, you could make them in one graphical program but if not able this remark is not mandatory

- UV- Vis please separate the spectra to make them one under the second (add more space - it will be easier to recognize them)

- In line 178 maybe the word "probably" will be better for the explanation

- line 177 what do you mean by "little difference" - did you mean shift?

- section starting from line 200 is a little bit strange - if you can not do something just point out the differences you don t have to confess. Just present what you did.

- line 229 please stop suggesting that some previous work hides something - rewrite the sentence and check the whole manuscript to avoid these statements. If something was not performed just present what you have done (see previous recommendations)

- figure 8 separate the spectra

- figure 10 should be improved separate the spectra and uniform them as previously recommended

- why did you not show the antiviral results?

Conclusions

- cross out for scientific honesty